# A Systematic Review on Healthcare Artificial Intelligent Conversational Agents for Chronic Conditions

**DOI:** 10.3390/s22072625

**Published:** 2022-03-29

**Authors:** Abdullah Bin Sawad, Bhuva Narayan, Ahlam Alnefaie, Ashwaq Maqbool, Indra Mckie, Jemma Smith, Berkan Yuksel, Deepak Puthal, Mukesh Prasad, A. Baki Kocaballi

**Affiliations:** 1School of Computer Science, Faculty of Engineering and IT, University of Technology Sydney, Sydney 2007, Australia; abdullahhatima.binsawad-1@student.uts.edu.au (A.B.S.); ahlam.alnefaie@student.uts.edu.au (A.A.); berkan.yuksel@student.uts.edu.au (B.Y.); mukesh.prasad@uts.edu.au (M.P.); baki.kocaballi@uts.edu.au (A.B.K.); 2School of Communication, Faculty of Arts and Social Sciences, University of Technology Sydney, Sydney 2007, Australia; bhuva.narayan@uts.edu.au (B.N.); indra.mckie@uts.edu.au (I.M.); 3School of Public Health, Faculty of Medicine and Health, The University of Sydney, Sydney 2007, Australia; amaq6135@uni.syndey.edu.au; 4School of Biomedical Engineering, Faculty of Engineering and IT, University of Technology Sydney, Sydney 2007, Australia; jemma.smith@student.uts.edu.au; 5Department of Electrical Engineering and Computer Science, Khalifa University, Abu Dhabi P.O. Box 127788, United Arab Emirates

**Keywords:** conversational agents, dialogue systems, relational agents, chatbot

## Abstract

This paper reviews different types of conversational agents used in health care for chronic conditions, examining their underlying communication technology, evaluation measures, and AI methods. A systematic search was performed in February 2021 on PubMed Medline, EMBASE, PsycINFO, CINAHL, Web of Science, and ACM Digital Library. Studies were included if they focused on consumers, caregivers, or healthcare professionals in the prevention, treatment, or rehabilitation of chronic diseases, involved conversational agents, and tested the system with human users. The search retrieved 1087 articles. Twenty-six studies met the inclusion criteria. Out of 26 conversational agents (CAs), 16 were chatbots, seven were embodied conversational agents (ECA), one was a conversational agent in a robot, and another was a relational agent. One agent was not specified. Based on this review, the overall acceptance of CAs by users for the self-management of their chronic conditions is promising. Users’ feedback shows helpfulness, satisfaction, and ease of use in more than half of included studies. Although many users in the studies appear to feel more comfortable with CAs, there is still a lack of reliable and comparable evidence to determine the efficacy of AI-enabled CAs for chronic health conditions due to the insufficient reporting of technical implementation details.

## 1. Introduction

The availability and use of conversational agents have been increasing due to advances in technologies such as natural language processing (NLP), voice recognition, and artificial intelligence (AI). Conversational agents (CAs), also known as chatbots or dialogue systems, are computer systems that communicate with users through natural language user interfaces involving images, text, and voice [1,2]. Google Assistance, Apple Siri, Amazon Alexa, and Microsoft Cortana are common CAs with voice-activated interfaces. In the last decade, CAs’ popularity has increased, particularly those that use unconstrained natural language [3,4,5]. For example, consumers can talk to CAs on their smartphones for daily tasks, such as managing their calendars and retrieving information [6,7].

Recently, AI-based CAs have demonstrated multiple benefits in many domains, especially in healthcare. It is used to deliver scalable, less costly medical support solutions that can help at any time via smartphone apps or online [8,9]. For example, support and follow-up for adults after cancer treatment via chatbot reduced the patients’ anxiety without needing a psychiatrist [10,11,12]. Hence, CAs can play an useful role in health care, improving consultations by assisting clinicians and patients, supporting consumers with behavior change, and assisting older people in their living environments [13,14,15]. They can also help in completing specific tasks such as self-monitoring and overcoming obstacles for self-management, which is important in chronic disease management and in the fight against pandemics [6,16].

Chronic conditions and mental health conditions are increasing worldwide. Chronic diseases are one of the biggest healthcare challenges of the 21st century [17,18]. Chronic conditions are “characterized by their long-lasting and persistent effects. Once present, they often persist throughout a person’s life, so there is generally a need for long-term management by individuals and health professionals” [1]. Additionally, chronic conditions reduce one’s quality of life and increase healthcare expenses through disability, repeated hospitalization, and treatment procedures. According to the World Health Organization statistics of 2020, non-communicable diseases (e.g., hypertension, diabetes, and depression) and suicide are still prevalent reasons for death in 2016 [19]. In the US, about 60% of adults have chronic diseases, causing the annual health care expenditure approximately 86.2% of the $2.6 trillion [20]. In 2018, the Australian Institute of Health and Welfare claims that diabetes is one of Australia’s eight common chronic conditions, contributing to 61% of the disease burden, 37% of hospitalizations, and 87% of deaths [21]. There are about 1.13 billion people who had suffered from hypertension in 2015, and the number is still increasing. About 46% of adults do not know that they have hypertension. All statistics about chronic conditions show how serious they are and their effect on people’s lives [19].

Some research studies have shown advantages from the use of AI-enabled CAs in different healthcare settings, such as enabling behavior change, coaching to support a healthy lifestyle, helping breast cancer patients, and self-anamnesis for therapy patients [7,22,23]. Prior systematic literature reviews explored a variety of CAs in general health care [1,6,24] and aspects of the personalization of health care chatbots using AI [25]. However, there is little evidence on the use of AI-based CAs in chronic disease health care. This paper aims to address the gap by reviewing different kinds of CAs used in health care for chronic conditions, different types of communication technology, evaluation measures of CAs, and AI methods used.

Section 2 presents methods explaining the search strategy, eligibility criteria, screening, and data extraction processes. Section 3 addresses the results that include descriptions of included studies, CAs, AI methods, and evaluation measures. Section 4 provides a discussion of findings and outcomes. Section 5 presents the conclusion and future work.

## 2. Methods

Reporting standards

A systematic literature review has been performed which followed the Preferred Reporting Items for Systematic Reviews and Meta-Analyses (PRISMA) checklist [24]. The review protocol in Appendix A was registered on OSF Preregistration, with DOI 10.17605/OSF.IO/GDWSH.

Search strategy

A systematic search was performed in February 2021, on PubMed Medline, EMBASE, PsycINFO, CINAHL, Web of Science, and ACM Digital Library, not restricted by year or language. Search terms included “conversational agents”, “dialogue systems”, “relational agents”, and “chatbots” (complete search strategy available in Appendix A) [1,6,25,26]. Gray literature that was also identified in those databases (including conference proceedings, theses, dissertations), were included for screening.

Study selection criteria

The criteria included primary research studies that focused on consumers, caregivers, or healthcare professionals in the prevention, treatment, or rehabilitation of chronic diseases using CAs, and tested the system with human users. Reviews, perspectives, opinion papers, or news articles were excluded based on exclusion criteria. In addition, studies that reported on evaluations based on human users interacting with the entire health system were excluded. The studies that evaluated only individual components of natural language understanding and CAs’ automatic speech recognition, dialogue management, response generation, and text-to-speech synthesis were excluded. The last exclusion criteria were studies using “Wizard of Oz” methods, where dialogue generated by a human operator rather than the CAs, were excluded [1,6,9].

Screening, data extraction, and synthesis

All references identified through the searches were downloaded. Then, duplicates were eliminated using reference managers (Endnote and Mendeley). Next, the titles and abstracts for each paper were exported from the reference manager into an Excel spreadsheet.

Before starting the screening process, the procedures of the screening were handled. After that, the first filter used was a screening filter based on the information contained in their titles and abstracts. Two independent reviewers conducted this screening. Two independent reviewers also conducted the full-text screening. The exclusion of an article was resolved with a Zoom meeting between two independent reviewers. Four reviewers extracted the following data for each study: first author, year of publication, study location, chronic condition, study aim, study types and methods, participants’ characteristics, evaluation measures, and main findings (Table 1). Evaluation measures were extracted based on three types: technical performance, user experience, and health-related measures. The technical performance of CAs was considered an objective assessment of the technical properties of the whole system. The technical performance measure is not included in Table 1 because most papers had not reported it, so the information about this measure will be in the next sections. User experience evaluation included the subjective assessment, where users tested the system properties or components based on their perspectives, via quantitative or qualitative methods [27]. Health-related measures were considered, alongside any health outcomes present in the included studies, such as diagnostic accuracy or symptom reduction.

Table 2 has retained the characteristics of the CAs (the categories defined in Box 1) that were evaluated in the included studies. In addition, it shows AI methods used, based on a list of keywords, defined from three systematic literature reviews for CAs in health care [1,6,27].

## 3. Results

The six databases that were searched retrieved 1754 articles. Then, the duplicates were removed, which resulted in 1087 unique articles. After the abstract and title screening, 110 articles remained. After the full-text screening, 80 of these were excluded. Thirty articles were considered eligible for inclusion in the systematic literature review. Four more papers were excluded during extraction data based on the exclusion criteria. Twenty-six articles were considered eligible for inclusion in the systematic literature review (Figure 1).

### 3.1. Description of Included Studies

The complete list of included studies (26 studies) used CAs to support tasks undertaken by patients (*n* = 14), clinicians (*n* = 1), and both (*n* = 11). Fourteen studies focused on patients mostly supporting education and self-care [28,29,30,31,32,33,34,35,36,37,38,39,40,41]. One study focused on clinicians, where the CAs were used to educate and increase awareness of the introductory psychology students in mental and physical health [42]. Eleven further studies supported patients and clinicians, where seven used CAs in treatment, education, and data collection [43,44,45,46,47,48,49]. Two studies helped in following-up [50,51], and two studies promoted medication adherence [52,53]. The most common chronic condition was diabetes (*n* = 5). Two studies were for type 1 [29,39], one study was for type 2 [28], and one focused on diabetes in general, whether the patients have type 1 or 2 [40]. One study focused on prediabetes with obesity [51]. The other four studies concentrated on different aspects of depression, such as symptoms and disorders [37,45,46,52]. Three studies dealt with cancer (breast cancer, geriatric oncology, after cancer treatment) [34,47,50]. Anxiety that could lead to depression [32,48], mental health [33,42], and schizophrenia [30,44] were the focus of two studies for each condition. Other conditions included hypertension [43], heart failure [38], HIV [53,54,55,56], long-term conditions [35], chronic pain [36], chronic problems in oral health [41], urinary incontinence [31], and post-traumatic stress disorder [49]. In terms of methods, the methods used in most studies were RCT, pilot study, or quasi-experimental. Only two studies had used mixed methods, and the remaining studies used other methods.

### 3.2. Description of Conversational Agents and AI Methods

Different technologies have supported CAs, including independent platforms, apps delivered via web or mobile device, short message services (SMS), and telephone (Table 2). Out of 26 conversational agents, 16 were chatbots (a computer program that simulates human conversation via voice or text communication). Seven were embodied conversational agents (ECA), a virtual agent that appeared on computer screens and was equipped with a virtual, human-like body that had real-time conversations with humans. One was a conversational agent in a robot, and another was a relational agent explicitly designed to remember history and manage future expectations in their interactions with users. One agent was not specified [43]. The characterisation of conversational agents are as shown in Table 3, and this summarization is adapted from Laranjo et al. 2018 [27].

The CAs in the papers used various AI methods such as speech recognition, facial recognition, and NLP. However, most studies did not provide sufficient information on the implementation details. In order to identify the AI methods, a list of common words (Appendix B) used for building AI CAs [1,6,27] were employed. Several papers reported that AI methods could improve the user’s interaction with the system [1,2,5,6,27]. For example, speech recognition can capture speech much faster than you can type. Half of the included papers utilized speech recognition in many CAs (e.g., chatbot, ECA, or relational agent). Although having speech recognition can capture speech much faster than typing, it could lead to difficulties with some keywords because of misinterpretation of words. Six studies did not report these technical methods.

### 3.3. Evaluation Measures

Evaluation measures were identified based on three types: technical performance (six studies), user experience (25 studies), and health-related measures (18 studies). The most common technical performance measures were accuracy (89–99.2% for five CAs) [31,37,40,43,48] and specificity (93–99.7% for three CAs) [37,40,46]. One study for hypertension identified that the rate of the achieved goal for the CAs was 96%. In addition, the authors clarified that the accuracy of the spoken dialogue system in cough and compliance were 81% and 41%, respectively [43]. Another study in glaucoma and diabetic conditions used Cohen’s D to calculate the task completion (k = 0.848), and the accuracy of the CAs was 89% [40]. Two studies were on depression (symptoms and major depressive disorder) and used finite-state dialogue management. The study for symptoms of depression noted that the written chatbot showed an accuracy of 99.2% and a specificity of 99.7% [52]. The spoken system for a major depressive disorder used embodied CAs, and showed sensitivity (49%) and specificity (93%) [46]. Two studies were about treating urinary incontinence and various chronic conditions such as pain and anxiety. The urinary incontinence article mentioned accuracy, but without clarifying the percentage or rate of accuracy [31]. Another paper for various chronic conditions (pain, anxiety, and depression) showed almost 92% accuracy in the breathing rate for patients [48].

Almost all studies reported on user experience except one study [43]. Helpfulness, satisfaction, and ease of use were the common features in more than half of the included studies. Three studies mentioned that users were unsatisfied. In two studies, the participants found the CAs hard to use. Regarding diabetes–type 2 [28], a study reported the feedback from patients through various measures, such as competency (85%), helpfulness and friendliness (86%). On the other hand, some patients described the embodied CAs as annoying (39%) and boring (30%). Another study for diabetes [40] illustrated the user experience through attractiveness (0.74), perspicuity (0.67), and efficiency (0.77), by using the scale of Cronbach’s Alpha Coefficient correlation. In mental health, a study for treatment and education reported that some users felt the chatbot was hard to engage with and had no availability to ask questions [33]. A study after cancer treatment clarified that the users found the chatbot nonjudgmental and helpful. Additionally, users supported recommending it to a friend (69%).

Regarding health-related measures, 18 out of the 26 studies included the health-related measures. The most common method that has been used is quasi-experimental, where it was used in 12 out of 26 studies. The second most common method used was RCT, used in six studies. One of the quasi-experimental studies evaluated the medication adherence system of interaction dialogue, finding decreased delirium (*p* < 0.001) and loneliness (*p* = 0.01) [45]. Another study showed a reduction in depression and anxiety by *p* = 0.053 and 0.029, respectively [32]. One RCT measured the outcomes using a 5-point Likert-type scale, finding improved self-management for older people with the chatbot (*p* = 0.001) [28]. One study reported quasi-experimental and RCT which evaluated a medication adherence intervention, finding that system use, medication adherence, physical activity, and satisfaction measures were high (84–89%) [44].

## 4. Discussion

The most commonly used method in the included studies was quasi-experimental, which was used in almost half of the included papers. This is aligned with the findings of the previous systematic reviews of CAs in healthcare [1,27]. Quasi-experimental demonstrates the involvement of real-world interventions, instead of artificial laboratory settings. It allows the research to move with higher internal validity than other non-experimental types of research. In addition, quasi-experimental design requires fewer resources and is less expensive compared with RCT. This systematic review introduced a list of AI CAs in healthcare for chronic disease. It reflects the efficiency, acceptability, and usability of the AI CAs in the daily education of, and support for, chronic disease patients. Our review reflected this as most of the included studies were published after 2016 (21 papers). Most included studies evaluated task-oriented AI CAs (23 studies out of 26) that are used to assist patients and clinicians through specific processes. The majority of the included studies were focused entirely on designing, developing, or evaluating AI CAs that are specific to one chronic condition. This finding implies that AI CAs evolve to provide tailored support for specific chronic conditions, rather than general interventions for a broad range of chronic conditions.

The outcomes of the included studies were assessed on three measures: technical performance, user experience, and a health-related measure. There were only six studies that mentioned some technical details. Due to the lack of details reported on the technical implementation of AI methods, it was not possible to establish consistent relationships with the intervention used, disease areas, and measured outcomes. The evaluation measures of the identified AI-based CAs and their effects on the targeted chronic conditions were not unified and broad. This inconsistency shows the complexity of contrasting and comparing the current AI CAs. Regardless of some studies that showed the complexity in use and chatbot constraints (four studies), most studies reported satisfaction with agents and feeling more comfortable than continuous follow-ups with a doctor in the hospitals. User experience was the most commonly reported measure (25 studies). It reflects the positive effect and enhancement of the quality of life in most studies through AI CAs that help patients who suffer from a chronic condition. This systematic review found that most included studies focused on designing, developing, or evaluating AI CAs for a specific chronic condition. That resulted in more accuracy, a tailored interaction with patients, and enhanced interventions for a wide range of conditions. In dialogue management, nine studies used a mixed initiative, whereas most applied system initiatives. No study in the included studies targeted the dialogue management of agent-based interactions. Moreover, these studies do not contain CAs that can be used across other populations. No analysis is applied on a broad scale, especially for communities or countries that suffer a lot from managing chronic conditions due to the high demand on hospitals or the cost and effort of following up with doctors. Targeting this area will help many people deal with chronic conditions and live their lives, especially as the CAs’ supporting preventive measures can prove very effective.

Compared to prior reviews focused on AI CAs for healthcare, we found only two review studies that targeted AI CAs for chronic conditions, where one of them focused on voice-based CAs only. Those reviews did not differentiate between the type of CAs used besides the AI methods used in each study, so this review focused on investigating the different types of dialogue management with the AI method used in each study. This review also focused on technical descriptions of the CAs used. Clarifying the technical features of the AI CAs will help to choose the appropriate type of AI CAs. Regarding limitations, most studies did not include technical performance details, which makes replicability of the studies reviewed problematic. Another limitation of the reviewed literature is the heterogeneity and the prevalence of quasi-experimental studies. This suggests that this is still a nascent field.

## 5. Conclusions

Many studies in this review showed some positive evidence for the usefulness and usability of AI CAs to support the management of different chronic diseases. The overall acceptance of CAs by users for the self-management of their chronic conditions is promising. Users’ feedback shows helpfulness, satisfaction, and ease of use in more than half of the included studies. Although the users in many studies appear to feel more comfortable with CAs, there is still a lack of reliable and comparable evidence to determine the efficacy of AI-enabled CAs for chronic health conditions. This is mainly due to the insufficient reporting of technical implementation details. Future research studies should provide more detailed accounts of the technical aspects of the CAs used. This includes developing a comprehensive and clear taxonomy for the CAs in healthcare. More RCT studies are required to evaluate the efficacy of using AI CAs to manage chronic conditions. Safety aspects of CAs is still a neglected area, and needs to be included as part of core design considerations.

## Figures and Tables

**Figure 1 sensors-22-02625-f001:**
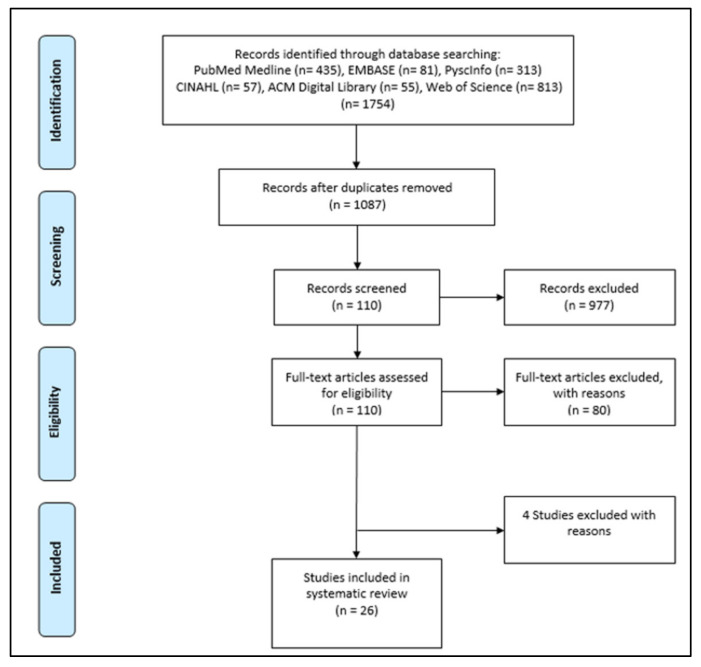
Flow Diagram.

**Table 1 sensors-22-02625-t001:** Overview and characteristics of included studies.

Author, Year	Study Location	Type of Chronic Condition	Study Aim	Study Type and Methods	Participants’ Characteristics	Evaluation Measures and Main Findings	
						User Experience	Health Related Measures
Azzini et al., 2003	Italy	Hypertension (patients with essential hypertension)	Data collection, developing a prototype home monitoring system.	**Quasi-experimental** (150 dialogues; 15 patients with essential hypertension entered the data at home; physicians used interface to store and update patient information).	Fifteen patients with no information about age, gender and duration.	Not reported	Not reported
Baptista et al., 2020	Australia–New South Wales, Queensland, Victoria, and Western Australia	Diabetes–Type 2 (T2D)	Self-management, education, and support.	**Qualitative** (six months baseline; 66 of the 93 patients completed a survey). **Quantitative** (between October 2017 to February 2018; 16 of the total patients had semi-structured interviews). **RCT** (testing the effectiveness of the T2D self-management smartphone app).	Ninety-three patients from My Diabetes Coach app. Sixty-six responses in 6 months post baseline. Nineteen of these respondents participated in the interviews. Avg. age: 57; male: 33; female: 33.	User experience feedback is as the following: helpful and friendly (86%), competent (85%), trustworthy (73%), likable (61%), not real (27%), boring (39%), annoying (30%), more motivated (44%), comfortable (36%), confident (21%), happy (17%), hopeful (12%), frustrated (20%), and feel guilty (17%).	Improving self-management. Older people have more interaction (*p* = 0.001). Participants who were interviewed showed more interactions with Laura (*p* = 0.001).
Beaudry et al., 2019	America-Vermont	Chronic condition (teenagers with pediatric Inflammatory Bowel Disease, Cardiology, or Type 1 Diabetes)	Learning self-care for teenagers (transition from pediatric to adult) with a chronic condition.	**Quasi-experimental** (24 weeks; pilot study on 13 teenagers with a chronic medical condition using a text messaging platform (chatbot) with scripted interactions)	Thirteen teenagers from the University of Vermont Children’s Hospital. Age: 14–17; duration: 24 weeks.	Participants agreed that overall, text messaging was the right channel for them, and the rate of one message per week was preferred. Avg. response for patients = 97%.	Participants suggest this chatbot should be expanded, and that it shows promise to help teenagers attain self-care skills on the transition journey.
Bickmore et al., 2010	America-Boston	Depressive Symptoms	Hospital patients know about their post-discharge self-care regimen through an automated system.	**Quasi-experimental** (one month; 131 patients interacted with the agent from their hospital beds; two rounds of pilot studies to assess usability, acceptance, and satisfaction with the agent; 347 subjects were enrolled and randomized; only 173 subjects were used into the relational agent of the study; nurses to follow up with patients).	One hundred and thirty patients from Boston Medical Centre. Age: 18; male: 70; female: 60; duration: 30 days.	All patients rated the agent with high satisfaction besides ease of use. Most patients (76%) preferred receiving discharge information from the agent instead of doctors or nurses.	Patients with major depressive symptoms showed more desire to continue with the agent (*p* < 0.05). The following are the attitude measures towards the agent: satisfaction: *p* = 0.37; usability: *p* = 0.49; continue *p* = 0.03; relationship: *p* = 0.30; preference: *p* = 0.29; adherence: *p* = 0.13.
Bickmore et al., 2010	America-Pennsylvania	Schizophrenia	Promoting antipsychotic medication adherence for patients with schizophrenia.	**Quasi-experimental** (initial range of responses, then modifying the list as needed to pilot testing). **RCT** (1–2 months; two RCTs. One with young adults (Bickmore et al., 2005 a, b) and another with geriatric patients (Bickmore et al., 2005a); both were conducted on home desktop computers).	Twenty patients from a mental health outpatient clinic. Age: 19–58; male: 67%; female: 33%; duration: 1–2 months. The nurse visited each participant’s home to explain how to use the computer and to make sure the software worked normally.	Sixteen participants out of the total completed the study of 1 month–daily use for ten min. The following are the participants’ ratings out of 5: trust: 4.4; liking: 4.3; satisfaction: 4.5; ease of use: 4.3; keep going with the system: 4.4.	Self-reported medication and physical activity adherence through all measures were very high (84–89%). Relationship with the agent was significantly correlated with system use (*p* < 0.05).
Bott et al., 2019	America-New York	Loneliness, Depression, Delirium, Falls	Supporting nurses and mitigating risks of hospitalization for elders.	**Quasi-experimental** (2 groups; group 1 (intervention)—41 participants received an avatar for the duration of their hospital stay, group 2—(control) 54 participants received a daily 15 min visit from a nursing student).	Ninety-five elders from an urban community hospital in New York. Age: over 65 years; male: 43; female: 52; the average length of stay for a patient: 3–6 days.	The mean for patient engagement data was as follows: number of check-ins: 71.30/day; observational and engagement time: 61 min/day; media files used: 11.50/day; completed protocol tasks: 6.5 tasks/day.	Delirium: intervention group: significant reduction (*p* < 0.001); control group: no change in the frequency. Loneliness: the intervention group experienced a decrease in loneliness compared with the control group (*p* = 0.01). Depression: no significant difference between the two groups (intervention and control). Falls: the fall rate reduced by 82% in the intervention group while rate increased in the control group.
Chaix et al., 2019	France and Europe	Breast Cancer	Support, education, and improving medication adherence.	**Analysis** (1 year; 4737 patients, collecting data to analyze the number of conversations between patients and chatbot) + **Prospective study** (8 months; 958 patients received a weekly survey).	Analysis for the conversations between patients and chatbot (Vik). Patients: 4737; male: 526; female: 4211; avg age: 48; duration: 1 year. Prospective study patients: 958; duration: 8 months; no details about gender.	The satisfaction rate was very high, 93.95% (900/958). Vik chatbot helpful and supported by 88% (943/958).	Not reported
Dworkin et al., 2018	America- Chicago	HIV	Promoting HIV medication adherence and retention in care.	**Iterative approach** (five months; 16 men; five iterative focus groups to develop the phone app, each group have 3–4 participants; participants were divided based on the questionnaire they filled out).	Sixteen men participated (African American men who have sex with men) recruited from four Universities of Illinois at Chicago. Age: 18–34; duration: January to May 2016.	All participants welcomed positive messages and images, while some participants did not welcome the negative messages and images. Participants liked the interaction with the instructional avatar. The first four focus groups showed that stigma emerged as a critical issue, but there were no concerns by the fifth group. Avatar was acceptable by almost all participants, except four, who hoped for an option to choose a female version.	The study revealed that stigma at different levels should be considered. Negative images can overwhelm participants and make them want to turn off the app and not return to it.
Easton et al., 2019	UK	Patients with an Exempla r Long-Term Condition (LTC; Chronic Pulmonary Obstructive Disease (COPD))	Data collection, support, self-management, and diagnosis.	**Co-design workshop** (10 patients; 2 co-design workshops including health professionals and patients to fill out questionnaires).	Ten patients were identified through the local British Lung Foundation Breathe Easy support group. Avg. age: 71; male: 5; female: 5. Workshop 1 was run in July 2017 and lasted 5 h. Workshop 2 was run in October 2017 and lasted 5 h.	Almost half participants strongly agreed to use this system frequently. Easy to use: 88%. Needing technical support: 50%.	Not reported
Greer et al., 2019	America	After Cancer Treatment	Support and follow-up	**RCT** (8 weeks; 45 young adults; 2 groups, group 1 was experimental group (25 young adults), group 2 was control group (20 young adults); all participants filling-out a survey at baseline).	Forty-five young adults from Facebook advertising, survivorship organizations and direct email. Age: 18–29; male: 9; female: 36; duration: 8 weeks.	The feedback of the chatbot is nonjudgmental. The chatbot was helpful: 64%. Recommend it to a friend: 69%.	After 4 weeks, participants in the experimental group reported an average reduction in anxiety of 2.58 standardized t-score units. Mixed-effects models revealed a trend-level (*p* = 0.09) interaction between group and time, with an effect size of 0.41. The experimental group also experienced greater reductions in anxiety when they engaged in more sessions (*p* = 0.06). There were no significant effects by group on changes in depression, positive emotion, or negative emotion.
Hauser-Ulrich et al., 2019	German and Swiss	Self-Management of Chronic Pain	Pain self-management	**RCT** (8 weeks; 102 participants were recruited online, 59 of them were in the intervention group (cognitive behaviour therapy), and the rest were in the control group are not related to pain management).	One hundred and two participants from the SELMA app. Avg. age: 43.7 years; male: 14; female: 88; duration: 2 months.	Participants mentioned the app was useful and easy to use. The avg. answer ratio of participants in the intervention group: 0.71.	In relation to impairment and pain intensity, the intention to change behavior was positive (*p* = 0.01). Compared with the control group, The intervention group did not show a significant change in pain-related impairment (*p* = 0.01).
Inkster et al., 2018	America- Brooklyn and Chicago	Symptoms of Depression	Data collection and self-reported symptoms of depression	**Quasi-experimental** (11 July 2017, and 5 September 2017; 129 users were divided into two groups (high users and low users); quantitative was to check the impact of the intervention; qualitative was to check the user experience with Wysa app).	One hundred and twenty-nine users from the Wysa app (high users, *n* = 108; low users, *n* = 21). No. of female and male: not reported; duration: 11 July 2017 to 5 September 2017.	Seventy-five users found the app favorable (82%). Thirteen users claimed that the app does not understand or repeat itself (14%). Seventy-five users commented that the app is not helpful (82%). The high users’ group had a highly significant improvement average (*p* < 0.001) compared with the low users’ group (*p* = 0.01).	Not reported
Lobo et al., 2017	Portugal	Heart Failure Care and Pharmacological Information	Managing information about medicines and increasing adherence	**Survey** (11 adults; participants filled out a questionnaire to assess system’s performance, feasibility, and drawbacks).	Eleven native Portuguese adults. Age: 22–33; no information about gender and duration.	System naturalness, information quality, and coherence scores were consistent among participants. Participants need an initial time on the system to know how it worked. The majority of the participants faced difficulties with the speech recognition of some keywords.	CARMIE has proven the capability of addressing the pharmacological and treatment information for heart failure daily care.
Neerincx et al., 2019	Netherlands and Italy	Diabetes–Type 1 (T1DM)	Support and manage children diabetes	**Iterative refinement process** (6 months; this process went through three cycles that include knowledge base, interaction, and some functions to achieve an effective partner for diabetes management).	Children from diabetes camps and hospitals in Netherlands and Italy. Age: 7–14; duration: 6 months.	Cycle1: Children have increased knowledge of T1DM. Children like the PAL actor (robot and its avatar). Children experience diabetes-related activities more positively. Cycle2: Children bond with the PAL actor via the robot and its avatar. Children are motivated to work on their personal objectives with PAL. >Cycle3: Children have increased situated knowledge on T1DM. Children are aware of the T1DM state and causes and develop self-efficacy. Children have a higher Quality of Life concerning T1DM. Children seamlessly follow the culture and hospital-dependent diabetes management processes. Children pursue relatively difficult goals.	Children in the intervention groups had a stronger increase in self-care score (*p* = 0.01). No effect on diabetes related quality of life in children (*p* = 0.02).
Rehman et al., 2020	Korea	Glaucoma and Diabetic Conditions	Data collection and diagnosing	**Experimental method** (60 min per patient; 11 groups based on availability and feasibility (three patients per group); each patient interacted with the chatbot individually) (119 responses from 11 countries (overseas students) for the questionnaire were sent by email from the university)).	Thirty-three international students from the University of Kyung Hee. Age: 18–43; male: 20; female: 13; 60 min per patient.	Using Cronbach’s Alpha Coefficient correlation of items per scale: attractiveness: 0.74; perspicuity: 0.67; efficiency: 0.77; dependability: 0.60; stimulation: 0.67; novelty: 0.48.	Not reported
Stephens et al., 2019	America- Boston	Obesity and Prediabetes	Self-reported progress, support and follow-up with a clinician	**Feasibility study** (6 months; 23 youth encouraged to use Tess chatbot to help users to achieve the progress).	Twenty-three youths with obesity symptoms from children’s healthcare system. Age: 9.78–18.54; male: 10; female: 13; duration: 6 months.	Ninety-six percent of the total patients reported this chatbot is helpful.	Not reported
O’Hara et al., 2008	America	Intellectual Disabilities; Poor Dental Hygiene	Education and self-management	**Quasi-experimental** (6 months; 36 dental patients used personal assistive devices (PDs) and had their oral health tracked by a dentist).	Thirty-six participants from a single dental practice. No information about age and gender; 9 participants left study partway through; duration: 6 months.	More than half of participants reported PDAs not functioning correctly (mostly problems keeping the battery charged).	Ten participants (40%) achieved improvement in at least three areas of oral health.
Philip et al., 2017	France	Major Depressive Disorders (MDD)	**Clinical interview** (major depressive disorder diagnosis)	Clinical interviews (179 participants with major depressive disorders; interview 1 with CA, interview 2 with sleep clinic psychiatrist).	One hundred and seventy-nine outpatients from a sleep clinic in Bordeaux University Hospital. Age: 18–65; male: 42.5%; female: 57.5%; duration: November 2014 to June 2015.	Acceptability was good—25.4 (E-scale 0–30). Seventy-three percent of patients scored above 24.	Not reported
Piau et al., 2019	France	Cancer (Geriatric Oncology)	Data collection	**Quasi-experimental** (7 weeks; 9 participants to test semi-automated CA).	Nine participants (undergoing chemotherapy after cancer diagnosis). Age: +65; male: 5; female: 4; duration: 6 months.	Ninety-seven percent satisfaction on the chatbot overall. Eighty-seven percent considered monitoring Useful. Eighty percent satisfied with monitoring frequency. Most valuable benefits were moral support 44% and treatment management 40%.	Not reported
Puskar et al., 2011	America	Schizophrenia	Treatment, support and education.	**Quasi-experimental** (1 month; 17 participants from a local outpatient clinic given laptop computers with a relational agent)	Seventeen patients completed the study, but only results from two participants were mentioned in the study. Age: 18–55; the majority is female; duration: 1 month.	Patient 1: Mr. Z. Found the tracking system easy and simple to use. Patient 5: Ms. Q. Noted that Laura asks too many questions, then became okay with this after more explanation. After the trial period, Ms. Q understood her illness and the importance of taking meds and felt Laura was “on her side”. In addition, the ability to choose the time of day gave her freedom.	Before CA, the participants had an adherence level of 21%, but with the CA, the rate rose to 46%.
Richards and Caldwell, 2018	Australia	Urinary Incontinence	Treatment and education	**Quasi-experimental** ((Pilot studies 1,2,3; 62 patients used web-based eADVICE service (without an added ECA) prior to consultation, with a specialist; study 1—10 patients (2012), study 2—25 patients (2013), study 3—27 patients (2014)) (pilot study 4; 13 patients; testing initial reactions to an ECA called “Dr Evie”. (pilot study 5; over 6 months; 29 participants tested usability and usefulness of eADVICE service + Dr Evie and patient adherence)).	Children with urinary incontinence. Age: 6–16; 79 families enrolled; 74 completed pre-study survey; males: 44; females: 30; duration: not reported.	Fifty-one (69.9%) parents and 45 (61.5%) of children were happy with the treatment. Satisfaction results—11 questions asked—all mean results between 2.37–3.36 on 5-point scale (1 = seldom, 5 = always). Usability results—nine questions asked—all mean results between 2.66–3.13 on a 4-point Likert scale (1 = strongly disagree, 4 = strongly agree, no neutral options).	Fifty-four participants (74%) reported being dry during the day for 14 consecutive days, and 19 participants (26%) reported being dry during the night for 14 consecutive days. Thirty-seven percent of the total participants reported decreased severity of symptoms.
Ryu et al., 2020	South Korea	Mental Health (Depression and Anxiety)	Treatment	**Quasi-experimental** Initial field study (1 day; 24 older adults; 10 min of use, video recording hands and screen; thematic analysis of interviews to find five features). Beta-testing field study (2 weeks; 25 older participants; 4 excluded from analysis; chat-initiated message three times a day; Epidemiologic Studies Depression and Beck Anxiety Inventory scales used before and after testing; negative polarity analysis of chat).	Initial testing had 24 older adults. male: 7; female: 17. Beta-testing had 25 older adults; 4 excluded from analysis for missing second interview; 4 lost their chat history, 2 declined chat history collection; no information about age and duration.	Most users preferred text-based interaction. Users that experienced technological usability issues preferred voice interaction. Eighty-three percent of users were positive about cognition-enhancing games.	The study reported a reduction in depression (*p* = 0.053), anxiety (*p* = 0.029), and negative polarity (*p* = 0.275) from pre- to post-study surveys.
Schroeder et al., 2018	America	Mental Health	Treatment and education	**Quasi-experimental** (4 weeks; 73 individuals; surveys containing OASIS and PHQ-9 scales for anxiety and depression, and 5-point Likert scale question for user satisfaction).	Seventy-three participants. Female: 65; male: 7; age: 18–63; duration: 4 weeks.	Users trusted the chatbot as it was based on Dr Marsha Linehan’s material on DBT. Some felt the chatbot was hard to engage with, too generic and impersonal. The reminders by the app helped the participants stay engaged. Some participants felt their learning was hindered by their inability to ask questions.	The study showed a significant reduction of depression (PHQ-9, linear and quadratic *p* < 0.001) and anxiety (OASIS, linear *p* < 0.001 and quadratic *p* < 0.05) between the intake and week one surveys. Participants saw a reduction in dysfunctional coping (*p* < 0.001), blaming others (*p* < 0.05) and an increase of DBT skills (*p* < 0.001) from intake to exit surveys.
Sebastian & Richards, 2017	Australia	Mental and Physical Health (Anorexia Nervosa)	Education and increased awareness	**RCT** (245 participants; 4 min video, variant-time ECA interaction, but same transcript length; 4-way design Mental health literacy (MHL) framework is used to assess stigma amongst participants).	Two hundred and forty-five undergraduate university students. Age: +18; no information about gender and study duration.	Participants improved their ability to recognize Anorexia Nervosa over time (*p* < 0.001). Participants less likely to recognize Anorexia Nervosa as another form of eating disorder over time (*p* < 0.001) and identify Anorexia Nervosa via non-psychiatric labels over time (*p* < 0.001).	Participants were found to have improved recognition of Anorexia Nervosa as a mental illness over time (*p* < 0.001). Both positive and negative volitional stigma was significantly lower in post-intervention 1 (*p* < 0.001) and significantly lower once again in post-intervention 2 (*p* < 0.001). Education strategies for females reduce traditional stigma in females (*p* < 0.05).
Shamekhi & Bickmore, 2018	America	Various Chronic Conditions; Pain, Anxiety, and Depression	Treatment and coaching	**RCT (Respiration)** (2 × 12 min meditations; Mindful Attention Awareness Scale (MAAS) used to assess mindfulness; the control group was given agent treatment without respiratory sensors). **RCT (Comparison)** (24 participants; 2 × 12 min meditations; the control group was shown Eckhart Tolle video; agent dialogue was modified to match the video).	Respiratory RCT had 21 participants. Age: +18; male: 38%; female: 62%; duration: 2 sessions; (12 min per session). Comparison RCT had 24 participants; Age: +18; male: 63%; female: 37%; duration: 2 sessions; (12 min per session).	Eighty-six percent of participants found the agent was more accurate with respiratory sensors. Many participants preferred breathing over tapping a screen as it was less distracting. Most participants found the synthetic voice acceptable, some enjoyed it, and few found it robotic. Most participants found the appearance of the agent calming. Most participants became more aware of their breathing after agent feedback.	The following shows the evaluation results: Instructor: Sensor (*p* = 0.32), video (*p* = 0.01); Meditation experience: Sensor (*p* = 0.03), video (*p* = 0.41); Interactivity: Sensor (*p* = 0.001), video (*p* = 0.02).
Tielman et al., 2017	Netherla-nds	Post-Traumatic Stress Disorder (PTSD).	Treatment.	**Quasi-experimental** (4 participants; 12 sessions with a CA to create a virtual diary, then the PTSD environment was recreated in Worldbuilder. Participants started with self-assessments, and sessions are closed with a questionnaire (5 pt. Likert scale)).	Four participants. Two males were war-veterans; two females experienced childhood sexual abuse; no information about age and study duration.	Participants rated the system usability scale between 73 and 75. The participants found the system useful (*p* = 0.001). There was significant variance in the results (95%). One participant found the agent and the instructional video to be detrimental to therapy.	On average, the questions provided helped the participants recall memory (*p* = 0.002). No difference found in system components (*p* = 0.30); participants rated the system well usable (*p* = 0.014). Average usefulness of questions: SD = 8.37. Average usefulness of system: SD = 16.17. No difference in components: Usefulness SD = 6.12, usability SD = 14.96.

Abbreviations: Avg.: average; OOV: out of vocabulary; RCT: randomized control trial, app: application; *p*: *p*-value; Laura: chatbot prototype who blends speech recognition, AI, and realistic 3D animation; SELMA app: A Digital Coach for Self-Management of Pain; Wysa app: AI-powered mental health app; CARMIE: a smartphone-based assistant developed with the aim to deliver information and knowledge-based advice to help chronic disease patients; CA: conversational agents; ECA: Embodied Conversational Agent; PDA: Personal digital assistance; eADVICE: electronic Advice and Diagnosis via the Internet following Computerised Evaluation; Dr. Evie: eVirtual agent for incontinence and enuresis; PHQ-9: Patient Health Questionnaire 9-item scale, measures the frequency and severity of depressive symptoms; DBT: Dialectical Behavior Therapy. OASIS: Development and Validation of an Overall Anxiety Severity and Impairment Scale; SD: standard deviation.

**Table 2 sensors-22-02625-t002:** Characteristics of the conversational agents evaluated in the included studies.

Author, Year	Type of Communication Technology; Type of Conversational Agent	AI Methods Used	Dialogue Management	Dialogue Initiative	Input	Output	Task-Oriented
Azzini et al., 2003	Smartphone and web-based; spoken dialog system.	Speech recognition and spoken dialog system.	Finite-state	Mixed	Spoken	Spoken, written	Yes
Baptista et al., 2020	Smartphone app; ECA.	Speech recognition, natural language processing.	Finite-state	System	Spoken, visual	Spoken, written, visual	Yes
Beaudry et al., 2019	Text messaging platform; chatbot.	Machine learning, NLU, NLP, deep learning, speech recognition.	Finite-state	System	Written	Written	Yes
Bickmore et al., 2010	Framework; ECA.	Speech recognition, synthetic voice.	Finite-state	System	Spoken, visual	Spoken, written, visual	Yes
Bickmore et al., 2010	Home desktop software; animated agent and interaction dialogues.	Not reported.	Finite-state	System	Visual	Spoken, visual	Yes
Bott et al., 2019	Platform; ECA.	Text-to-speech, NLU.	Frame-based	Mixed	Spoken, visual	Spoken, written, visual	Yes
Chaix et al., 2019	Smartphone and web-based; chatbot.	Machine learning, NLP.	Finite-state	System	Written, visual	Written	Yes
Dworkin et al., 2018	Smartphone app; Avatar-based embodied agent.	Not reported.	Finite-state	Mixed	Spoken, written, visual	Spoken, written, visual	Yes
Easton et al., 2019	Web-based; avatar and chatbot.	NLP, speech recognition.	Frame-based	Mixed	Spoken, written	Spoken, written, visual	Yes
Greer et al., 2019	Facebook messenger; chatbot.	Not reported.	Finite-state	System	Written	Written, visual	Yes
Hauser-Ulrich et al., 2019	Smartphone app; chatbot.	Not reported.	Finite-state	System	Written	Written, visual	No
Inkster et al., 2018	Smartphone app; chatbot.	Machine learning, unsupervised learning.	Finite-state	System	Written	Written, visual	Yes
Lobo et al., 2017	Android app; chatbot.	Speech recognition, speech synthesis, spoken natural language, hidden Markov model, natural language Understanding, natural language dialogue system.	Frame-based	Mixed	Spoken, written	Spoken, written	Yes
Neerincx et al., 2019	Platform independent app, robot and avatar.	Machine learning, deep learning, speech recognition, speech synthesis.	Finite-state	System	Visual	Spoken, written, visual	Yes
Rehman et al., 2020	Android app; chatbot.	NLU, speech recognition, text to speech synthesis, neural network algorithm, machine learning, natural language processing, deep learning, spoken dialog.	Frame-based	User	Spoken, written	Spoken, written	Yes
Stephens et al., 2019	SMS text messaging; chatbot.	Not reported.	Frame-based	Mixed	Written	Written	Yes
O’Hara et al., 2008	Personal Digital Assistants (PDAs).	Not reported.	Finite-state	System	Written	Written	Yes
Philip et al., 2017	Home desktop software; Virtual human ECA.	Speech recognition, synthetic voice.	Finite-state	System	Spoken	Spoken	Yes
Piau et al., 2019	Semi-automated smartphone messaging system; chatbot.	Speech to text.	Finite-state	System	Written	Written	Yes
Puskar et al., 2011	Home desktop software; Relational Agent.	NLU, facial recognition, speech dialogue system.	Frame-based	System	Written	Written	Yes
Richards and Caldwell, 2018	Website; Avatar and Empathic ECA ^a^.	Speech to text.	Finite-state	System	Visual	Written; spoken	No
Ryu et al., 2020	Smartphone app; chatbot.	Speech recognition.	Frame-based	System	Visual	Written	No
Schroeder et al., 2018	Smartphone app; chatbot.	Not reported.	Finite-state	System	Visual	Written	Yes
Sebastian & Richards, 2017	Platform independent app; ECA.	Not reported.	Finite-state	System	Visual	Written	Yes
Shamekhi & Bickmore, 2018	Home desktop software; an animated agent with spoken dialogue and sensing.	Spoken dialog system.	Frame-based	System	Respiration sensor	Spoken	Yes
Tielman et al., 2017	Home desktop software; an animated agent with spoken dialogue.	Spoken dialog system.	Finite-state	System	Visual	Spoken; written	Yes

Abbreviations: app: application; ECA: Embodied Conversational Agent; ^a^ Empathic ECA: empathic agent that provides face-to-face conversation in an empathic and caring way, to act as a virtual doctor for the family to interact with.

**Table 3 sensors-22-02625-t003:** Characterisation of conversational agents (Laranjo et al. 2018 [27]).

**Dialogue management**	Finite-state	The user is taken through a dialogue consisting of a sequence of pre-determined steps or states.
	Frame-based	The user is asked questions that enable the system to fill slots in a template in order to perform a task.
		The dialogue flow is not pre-determined, but it depends on the content of the user’s input and the information that the system has to elicit.
	Agent-based	These systems enable complex communication between the system, the user, and the application. There are many variants of agent-based systems, depending on what aspects of intelligent behavior are designed into the system. In agent-based systems, communication is viewed as the interaction between two agents, each of which is capable of reasoning its own actions and beliefs, and sometimes the actions and beliefs of the other agent. The dialogue model takes the preceding context into account, with the result that the dialogue evolves dynamically as a sequence of related steps that build on each other.
**Dialogue initiative**	User	The user leads the conversation.
	System	The system leads the conversation.
	Mixed	Both the user and the system can lead the conversation.
**Input modality**	Spoken	The user uses spoken language to interact with the system.
	Written	The user uses written language to interact with the system.
**Output modality**	Spoken, Written, visual (e.g., non-verbal communication like facial expressions or body movements).	
**Task-oriented**	Yes	The system is designed for a particular task and is set up to have short conversations, in order to get the necessary information to achieve the goal (e.g., booking a consultation).
	No	The system is not directed to the short-term achievement of a specific end-goal or task (e.g., purely conversational chatbots).

## Data Availability

Not applicable.

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
