# Peer review of "A Systematic Review on Healthcare Artificial Intelligent Conversational Agents for Chronic Conditions"

_sensors, 2022, doi:10.3390/s22072625_

Round 1
Reviewer 1 Report
I have the following recommendations regarding improvement of the paper.
- Include some other recent studies at least 10 published in the last 3~5 years.
- In the abstract the protocol registration can be mentioned as a footnote instead of writing it in the abstract.
- More details are required in the introduction section.
- Motivation of the study should be mentioned.
- Contributions and research gap from the previous studies should be included.
- Include a paragraph by the end of the introduction section, which describes the details of each section of the manuscript.
- The technical performance column in the Table 1 is “Not reported” for most of the studies. I recommend excluding this column and mention the study in the text for which this information is reported.
- Exclude the first column in the Table 1 i.e., the paper number.
- Exclude the first column in the Table 2 i.e., the paper number. Because it is mentioned in the second column which paper refers to the row of the table.
- If possible, include a graph i.e., chart, or plot which summarize the results
- In discussion, there is an uneven distribution of the paragraphs. Some are long while others are short.
- The conclusion is very short. i.e., just 5 lines almost with remaining lines discussing the future works. Add some details on the proposal and achievements in terms of result findings.
Author Response
Reviewer 1 Comment 1:
Include some other recent studies at least 10 published in the last 3~5 years.
Response
Thank you. We have 50 references. More than 30 references were cited in the manuscript published in this range of 3~5 years. We believe the paper sufficiently covers the recent relevant literature.
Reviewer 1 Comment 2:
In the abstract the protocol registration can be mentioned as a footnote instead of writing it in the abstract.
Response
Thank you. The protocol registration was removed from the abstract. It is written in Report standards under the Methods section.
Reviewer 1 Comment 3,4:
More details are required in the introduction section.
Motivation of the study should be mentioned.
Response
Thank you. We have extended the introduction by adding sentences that show how serious is chronic conditions as a part that shows the need.
According to the World Health Organisation statistics of 2020, non-communicable diseases (e.g., hypertension, diabetes, and depression) and suicide are still prevalent reasons for death in 2016 [19]. In the US, adults with chronic diseases are about 60%, causing the annual health care expenditures approximately 86.2% of the $2.6 trillion [20]. In 2018, the Australian Institute of Health and Welfare claims that diabetes is one of Australia's eight common chronic conditions contributing to 61% of the disease burden, 37% of hospitalisations, and 87% of deaths [21]. There are about 1.13 billion people who had suffered from hypertension in 2015, and the number is still increasing. About 46% of adults do not know that they have hypertension condition. All statistics about chronic conditions show how serious they are and their effect on people's lives [19].
Some research studies have shown the use of AI-enabled conversational agents in different healthcare settings, such as enabling behaviour change, coaching to support a healthy lifestyle, helping breast cancer patients, and self-anamnesis for therapy patients [7], [22], [23]. Prior systematic literature reviews explored a variety of conversational agents in health care in general [1], [6], [24] and aspects of personalisation of health care chatbots using AI [25]. However, there is little evidence on the use of AI-based conversational agents in chronic disease health care. This paper aims to address the gap by reviewing different AI conversational agents used in health care for chronic conditions, different types of communication technology, evaluation measures of conversational agents, and AI methods used.
Reviewer 1 Comment 5:
Contributions and research gap from the previous studies should be included.
Response
Thank you. We have added a few sentences in the introduction section. Besides, we mentioned a part of the discussion compared to the previous systematic reviews.
Prior systematic literature reviews explored a variety of conversational agents in health care in general [1], [6], [24] and aspects of personalisation of health care chatbots using AI [25]. However, there is little evidence on the use of AI-based conversational agents in chronic disease health care. This paper aims to address the gap by reviewing different AI conversational agents used in health care for chronic conditions, different types of communication technology, evaluation measures of conversational agents, and AI methods used.
Compared to prior works, reviews focused on AI conversational agents for healthcare, and we found only two review studies targeted AI conversational agents for chronic conditions, where one of them focused on voice-based only. Those reviews did not differentiate the type of conversational agents used besides the AI methods used in each study, so this review focused on investigating the different types of dialogue management with the AI method used in each study. This review also focused on technical descriptions of the CAs used. Clarifying the technical features of the AI conversational agents will help to choose the appropriate type of AI conversational agents.
Reviewer 1 Comment 6:
Include a paragraph by the end of the introduction section, which describes the details of each section of the manuscript.
Response
Thank you. Section 2 presents methods explaining the search strategy, eligibility criteria, screening, and data extraction processes. Section 3 addresses the results that include descriptions of included studies, conversational agents, AI methods, and evaluation measures. Section 4 provides a discussion of findings and outcomes. Section 5 presents the conclusion and future work.
Reviewer 1 Comment 7,8,9:
The technical performance column in the Table 1 is "Not reported" for most of the studies. I recommend excluding this column and mention the study in the text for which this information is reported.
Exclude the first column in the Table 1 i.e., the paper number.
Exclude the first column in the Table 2 i.e., the paper number. Because it is mentioned in the second column which paper refers to the row of the table.
Response
Thank you. we have revised the tables as suggested.
Reviewer 1 Comment 11:
In discussion, there is an uneven distribution of the paragraphs. Some are long while others are short.
Response
Thank you. We have revised and fixed the distribution of the paragraphs.
Reviewer 1 Comment 12:
The conclusion is very short. i.e., just 5 lines almost with remaining lines discussing the future works. Add some details on the proposal and achievements in terms of result findings.
Response
Thank you. We have revised and added some details in conclusion.
Although the users of many studies appear to feel more comfortable with conversational agents, there is still lack of reliable and comparable evidence to determine the efficacy of AI-enabled CAs for chronic health conditions. This is mainly due to the insufficient reporting of technical implementation details. Future research studies should provide more detailed accounts of technical aspects of the CAs used. This includes developing a comprehensive and clear taxonomy for the conversational agents in healthcare. More RCT studies are required to evaluate the efficacy of using AI CAs to manage chronic conditions. Safety aspects of CAs, still a neglected area, need to be included as part of core design considerations.
Reviewer 2 Report
This paper is a review of the AI conversational agents. The review was conducted in a systematic way and was quite extensive. I think that it could be accepted after a final English editing and minor corrections that authors find.
Author Response
Reviewer 2 Comment:
This paper is a review of the AI conversational agents. The review was conducted in a systematic way and was quite extensive. I think that it could be accepted after a final English editing and minor corrections that authors find.
Response
Thank you for your positive feedback. The revised paper has been proofread by the native speakers.
Reviewer 3 Report
Authors presents a manuscript titled "A Systematic Review on Healthcare Artificial Intelligent Con-versational Agents for Chronic Conditions".
The manuscript has been submitted to an special issue : "Advanced Machine Learning Techniques for Biomedical Imaging Sensing and Healthcare Applications"
Briefly, the manuscript deals with a meta-analysis of published papers related to the role of conversational agents in health care in chronic conditions.
If I have understood correctly the structure of the manuscript, the only reference to artificial intelligence throughout the manuscript was the Appendix B, composed of keywords related to artificial intelligence.
So without judging the scientific quality of the manuscript, I consider that the submitted manuscript is out of the scope and not suitable for publication in this special issue of Sensors Journal.
Author Response
Reviewer 3 Comment:
The manuscript has been submitted to an special issue : "Advanced Machine Learning Techniques for Biomedical Imaging Sensing and Healthcare Applications"
Thank .
Briefly, the manuscript deals with a meta-analysis of published papers related to the role of conversational agents in health care in chronic conditions.
If I have understood correctly the structure of the manuscript, the only reference to artificial intelligence throughout the manuscript was the Appendix B, composed of keywords related to artificial intelligence.
So without judging the scientific quality of the manuscript, I consider that the submitted manuscript is out of the scope and not suitable for publication in this special issue of Sensors Journal
Response
Thank you. Our review focused on the papers that used AI conversational agents for chronic conditions. Moreover, our paper examined AI methods used in the included studies in Table 2.
In the discussion section, we added the following paragraph.
Due to the lack of details reported on the technical implementation of AI methods, It was not possible to establish consistent relationships with the intervention used, disease areas, and measured outcomes. The evaluation measures of the identified AI-based conversational agents and their effects on the targeted chronic conditions were not unified and broad. This inconsistency shows the complexity of contrasting and comparing the current AI CAs.
Moreover, there were multiple statements on AI CAs without explicitly referring to them as AI CAs. We have revised the discussion section and added AI CAs in the places where needed.
Round 2
Reviewer 1 Report
My comments have been addressed well. Thanks
Reviewer 3 Report
Authors have improved introduction, results and discussion sections. It can be acccepted for publication as systematic review paper.